# CD21^lo^ B Cells Could Be a Potential Predictor of Immune-Related Adverse Events in Renal Cell Carcinoma

**DOI:** 10.3390/jpm12060888

**Published:** 2022-05-28

**Authors:** Kenichi Nishimura, Tatsuya Konishi, Toshiki Ochi, Ryuta Watanabe, Terutaka Noda, Tetsuya Fukumoto, Noriyoshi Miura, Yuki Miyauchi, Tadahiko Kikugawa, Katsuto Takenaka, Takashi Saika

**Affiliations:** 1Department of Urology, Ehime University Graduate School of Medicine, Toon 791-0295, Japan; watanabe.ryuta.cu@ehime-u.ac.jp (R.W.); noda.terutaka.tl@ehime-u.ac.jp (T.N.); fukumoto.tetsuya.it@ehime-u.ac.jp (T.F.); miura.noriyoshi.mk@ehime-u.ac.jp (N.M.); miyauchi.yuki.mf@ehime-u.ac.jp (Y.M.); kikugawa.tadahiko.my@ehime-u.ac.jp (T.K.); saika.takashi.ol@ehime-u.ac.jp (T.S.); 2Department of Hematology, Clinical Immunology and Infectious Diseases, Ehime University Graduate School of Medicine, Toon 791-0295, Japan; konishi.tatsuya.kq@ehime-u.ac.jp (T.K.); ochi.toshiki.eg@ehime-u.ac.jp (T.O.); takenaka.katsuto.hy@ehime-u.ac.jp (K.T.)

**Keywords:** immune checkpoint inhibitor, immune-related adverse events, CD21lo B cells, renal cell carcinoma, combination checkpoint blockade

## Abstract

Immune checkpoint inhibitor (ICI) therapy increases the risk of immune-related adverse events (irAEs). In particular, combination checkpoint blockade (CCB) targeting inhibitory CTLA-4 and PD-1 receptors could lead to irAEs at a higher rate than ICI monotherapy. Management of irAEs is important while using ICIs. However, there are no reliable biomarkers for predicting irAEs. The aim of this study was to elucidate early B cell changes after CCB therapy in patients with renal cell carcinoma (RCC) and verify whether B cells can be a predictor of irAEs. This prospective cohort study was conducted with 23 Japanese patients with metastatic RCC. An increase in the proportion of CD21^lo^ B cells and CD21^lo^ memory B cells was confirmed following CCB therapy. Although there were no differences in clinical outcomes between irAE and no-irAE groups, the proportion of CD21^lo^ B cells at baseline was lower in the irAE group, with a significant increase after the first cycle of CCB therapy. Further analysis revealed a moderate correlation between irAEs and CD21^lo^ B cell levels at baseline (area under the curve: 0.83, cut-off: 3.13%, sensitivity: 92.3, specificity: 70.0). This finding indicates that patients with low baseline CD21^lo^ B cell levels warrant closer monitoring for irAEs. The clinical registration number by the Certified Review Board of Ehime University is No. 1902011.

## 1. Introduction

With the development of immune checkpoint inhibitors (ICIs), cancer treatment has undergone a major paradigm shift. Combination checkpoint blockade (CCB) therapy with ipilimumab and nivolumab shows better response rates and progression-free survival than conventional targeted therapies in unresectable or metastatic renal cell carcinoma (RCC) [1]. However, unlike cytotoxic agents and molecular-targeted therapies, ICIs are associated with a risk of immune-related adverse events (irAEs). CCB therapy causes the development of irAEs at a higher rate than ICI monotherapy [1,2,3].

The prediction and management of irAEs are important in clinical practice. Biomarkers for early prediction of irAEs have been explored [4,5,6,7,8,9,10,11,12]. In case of non-organ specific biomarkers in CCB therapy, Das et al. reported that among patients with melanoma who received CCB therapy, the proportion of B cells in the peripheral blood decreases, whereas that of CD21^lo^ B cells increases, after treatment in patients who developed grade 3 or higher irAEs. This finding suggests that early changes in B cells following CCB therapy can be used to identify patients who are at increased risk of irAEs [4]. However, as the dose of CCB therapy is different for RCC and melanoma, the association between early B cell changes and irAEs in patients with RCC being treated with CCB therapy remains unclear. Since the incidence of irAEs varies depending on the dose of ICIs [3].

The aim of this study was to elucidate early B cell changes after CCB therapy in patients with RCC and verify whether B cells can be a predictor of irAEs.

## 2. Material and Methods

### 2.1. Study Protocol

Patients with metastatic RCC who were selected for CCB therapy were registered at the Ehime University Hospital, National Hospital Organization Shikoku Cancer Center, Ehime Prefectural Central Hospital, Uwajima City Hospital, and Matsuyama Red Cross Hospital from April 2019 to April 2021. Excluding criteria were patient age under 20 years, history of steroid treatment, ICI treatment, and autoimmune diseases. Follow-up period defines the period from the start of CCB therapy to the final observation date. IrAEs were assessed using Common Terminology Criteria for Adverse Events (CTCAE) ver 5.0. In Japan, the CCB regimen for metastatic RCC therapy comprises four doses of nivolumab (240 mg per body weight) combined with ipilimumab (1 mg per kg), administered intravenously every 3 weeks, followed by nivolumab (240 mg per body) every 2 weeks. Peripheral blood samples were collected at baseline and 3 and 6 weeks after the first cycle of CCB therapy. Based on previously published reports, human peripheral blood mononuclear cells (PBMCs) were isolated using Ficoll-Paque (GE Healthcare, Tokyo, Japan) and stored at −80 °C until use [5]. The samples were analysed using flow cytometry, and clinical characteristics were recorded.

This prospective cohort study was approved by the Certified Review Board of Ehime University (No. 1902011). All experiments were performed in accordance with the relevant guidelines and regulations. All subjects provided written informed consent prior to their participation in the study.

### 2.2. Flow Cytometry

PBMCs were obtained from the blood samples collected from all the patients. Then, approximately 2.0 × 10^5^ human PBMCs collected before and after CCB treatment were stained with fluorochrome-conjugated antibodies at the same time, and characteristics of human immune cells were compared. Briefly, fluorescein isothiocyanate (FITC)-anti-human CD21 (clone Bu32) monoclonal antibody (mAb), PE-anti-human PD-1 mAb (clone EH12.2H7), PC5-anti-human CD38 mAb (clone HIT2), APC-Cy7-anti-human CD27 mAb (clone M-T271), and BV421-anti-human CD19 mAb (clone HIB19) were used to analyse human B cells. CD19^+^CD27^+^ cells, CD19^+^CD27^−^ cells, and CD19^+^CD27^+^CD38^++^ cells were determined as memory B cells, naïve B cells, and plasmablasts, respectively. In addition, human T cells were analysed by staining with allophycocyanin (APC)-anti-human LAG-3 mAb (clone 7H2C65) along with PC5-anti-human CD8 mAb (clone B9.11), APC-Cy7-anti-human CD4 mAb (clone RPA-T4), and BV421-anti-human CD3 mAb (clone UCHT1). All samples were analysed using a Gallios flow cytometer (Beckman Coulter, Tokyo, Japan) and FlowJo ver.10.7.1 (Becton Dickinson, Tokyo, Japan).

### 2.3. Statistical Analysis

For variables with a non-normal distribution, data are presented as the median, and the groups were compared using the Mann–Whitney U test. Categorical variables were compared using the chi-square test. Consecutive variables (at baseline, 3, and 6 weeks after the first cycle of CCB therapy) were compared using the Wilcoxon signed-rank test. The proportion of B cell subsets between the irAE and no-irAE groups was analysed using the Mann-Whitney’s U test. A receiver operating characteristic curve was used to analyse the utility of the B cell subsets for predicting the risk of irAEs after CCB therapy. Significance was defined as *p* < 0.05 using a two-tailed test. Graphs were prepared, and statistical analysis was conducted using GraphPad Prism software, version 9 (MDF, Tokyo, Japam).

## 3. Result and Discussion

### 3.1. Clinical Characteristics

This study was a prospective cohort study in multiple institutions.

We enrolled 23 Japanese patients with metastatic RCC (18 males and 5 females) who were treated by CCB therapy from April 2019 to April 2021. The median follow-up period was 137 days (range: 23–562). The median age was 70 years (range: 45–87 years). IrAEs occurred in 13 cases (57%). The median onset time of irAEs was 42 days (range: 6–100 days). Grade 3 or higher irAEs occurred in seven cases. The main organs afflicted by irAEs were the lungs in four cases, thyroid gland in four cases, skin in two cases, and other organs in four cases. Comparison of the irAE and no-irAE groups revealed no significant difference in clinical characteristics (Table 1).

### 3.2. Changes in the Peripheral Blood after CCB Therapy in All Cases

Das et al. reported that the proportion of B cells in the peripheral blood decreases, whereas the proportion of CD21^lo^ B cells and plasmablasts increases, after CCB therapy [4].

We analysed changes in the peripheral blood after CCB therapy for metastatic RCC. No significant changes were observed in the proportion of circulating B cells at the baseline, 3, or 6 weeks (Figure 1a, n.s.). Further analysis showed that the proportion of CD21^lo^ B cells tended to increase from 3 weeks, and a significant increase was observed at 6 weeks (Figure 1b, BL vs. 6 w *p* < 0.01, 3 w vs. 6 w *p* = 0.01). The proportion of memory CD21^lo^ B cells significantly increased from 3 weeks (Figure 1e, BL vs. 3 w *p* < 0.01, BL vs. 6 w *p* < 0.001). Conversely, no significant changes were observed in the proportion of CD21^hi^ B cells, CD21^lo^ naïve B cells, and plasmablasts (Figure 1c,d,f, n.s.). The reason for the slower increase in CD21^lo^ B cells in our study compared to the study by Das et al. may be attributed to the difference in the ipilimumab dose. The recommended dose of ipilimumab for treating melanoma is 3 mg/kg, whereas that for treating renal cancer is 1 mg/kg. Ipilimumab, a CTLA-4 specific antibody, acts during the priming phase, in which antigen-presenting cells present cancer antigens to T cells in the lymph nodes and activate the T cells. Thus, the higher the dose of ipilimumab, the stronger the expected B cell response. Next, significantly decreased CD4^+^ T cell and increased CD8 ^+^ T cell proportions were observed 6 weeks after CCB therapy (Figure 1g, BL vs. 6 w *p* < 0.05; Figure 1h, BL vs. 6 w *p* < 0.01). The proportion of lymphocyte activation gene 3 expression (LAG-3) by CD4^+^ T cells increased significantly after CCB therapy (Figure 1i, BL vs. 3 w *p* < 0.01, BL vs. 6 w *p* < 0.01). LAG-3 CD4^+^ T cells increased by ICIs selectively activate Regulatory T cells to relate both anti-cancer effect and autoimmune disease [13]. In addition, LAG-3 CD4^+^ T cells bind to MHC class II molecules on the surface of antigen presenting cells (APCs) [14]. The APCs induce differentiation into Th1/Th2 cells. Th1 cells then activate B cells [15]. Therefore, ICIs may impact B cell function indirectly by exerting effects on T cells.

Eight patients who developed irAEs after 3 weeks of CCB therapy were analysed at baseline and 3 weeks after CCB therapy. The results of these eight patients were compared with those of patients in no-irAE group. No change in the proportion of B cells in PBMCs due to CCB therapy was observed in either of the groups (Figure 2a, n.s.). However, the proportion of CD21^lo^ B cells in the irAE group increased significantly 3 weeks after CCB therapy (Figure 2b, *p* < 0.05). Next, we evaluated whether CD21^lo^ B cell proportion at the baseline could predict irAEs. The proportion of CD21^lo^ B cells at the baseline was significantly lower in the irAE group (Figure 3a, *p* < 0.01). Moreover, the proportion of CD21^lo^ B cells at the baseline showed a moderate correlation with irAEs (Figure 3b, area under the curve: 0.83, cut-off: 3.13%, sensitivity: 0.92, specificity: 0.70).

CD21^lo^ B cells are considered anergic or exhausted B cell subsets [16,17]. The function of anergic B cells has not yet been clarified. Anergic B cells can bind to self-antigens but undergo intrinsic functional inactivation following an antigen encounter [18]. The disruption of B cell anergy is a potential mechanism underlying the onset of autoimmune disorders [19,20]. Therefore, CD21^lo^ B cells, as anergic B cells, are considered to be involved in the regulation of autoreactive cells. We hypothesised that the inability to control autoreactive immune cells in patients with a low proportion of CD21^lo^ B cells may lead to the development of irAEs. This indicates that the proportion of CD21^lo^ B cells at the baseline is a predictor of the onset of irAEs. Patients with low baseline CD21^lo^ B cell levels may require closer monitoring for irAEs.

The limitation of our study is that the number of enrolled patients was relatively small. Therefore, our finding will need to be validated in a larger cohort.

## Figures and Tables

**Figure 1 jpm-12-00888-f001:**
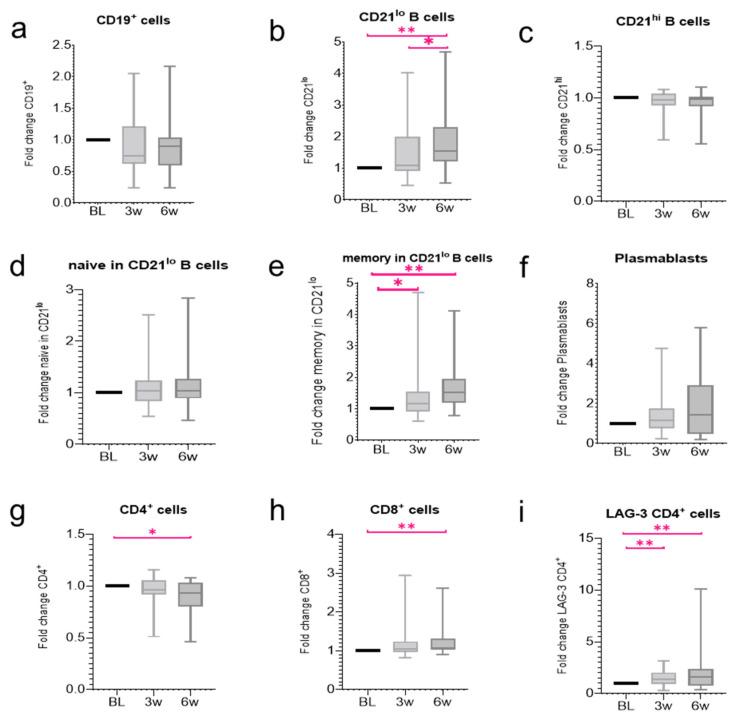
Changes in the proportion of the peripheral blood cells after combination checkpoint blockade (CCB) therapy. The box plot shows the fold change in the proportion of cells compared with baseline (BL) as well as 3 and 6 weeks after therapy. Changes in the proportion of (**a**) circulating B cells, (**b**) CD21^lo^ B cells, (**c**) CD21^hi^ B cells, (**d**) CD21^lo^ naïve B cells, (**e**) CD21^lo^ memory B cells, (**f**) plasmablasts, (**g**) circulating CD4^+^ cells, (**h**) CD8^+^ cells, and (**i**) LAG-3 CD4^+^ cells. All data represent the mean ± standard error of mean (SEM). * *p* < 0.05 and ** *p* < 0.01 by two-tailed Wilcoxon signed-rank test.

**Figure 2 jpm-12-00888-f002:**
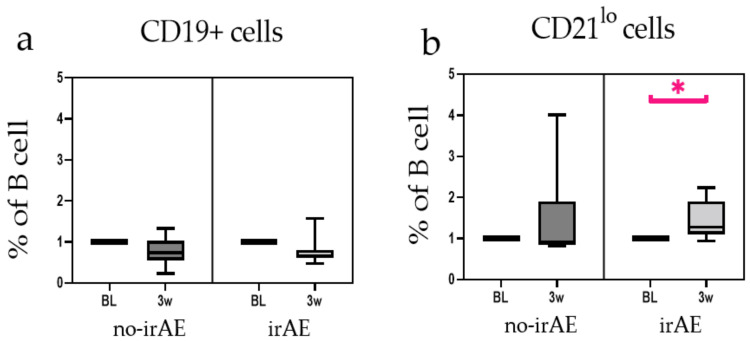
Immune-related adverse event (IrAE) group included eight patients who developed irAEs after 3 weeks. (**a**) Changes in proportions of circulating B cells are represented as the percentage of total peripheral blood mononuclear cells (PBMCs). (**b**) Changes in proportions of CD21^lo^ B cells. All data represent the mean ± standard error of mean (SEM). * *p* < 0.05 by two-tailed Wilcoxon signed-rank test.

**Figure 3 jpm-12-00888-f003:**
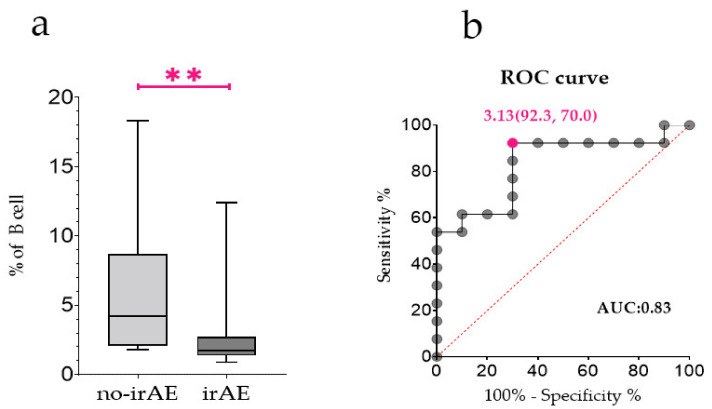
The proportion of CD21^lo^ B cells at the baseline. (**a**) Comparison between no-immune-related adverse event (irAE) group (*n* = 10) and irAE group (*n* = 13) for percentages of CD21^lo^ B cells among B cells at baseline. (**b**) Receiver operating characteristic curve showing utility of CD21^lo^ B cells at baseline for predicting the risk of irAEs in patients. ** *p* < 0.01 by two-tailed Wilcoxon signed-rank test.

**Table 1 jpm-12-00888-t001:** Comparison of clinical characteristics between no-immune-related adverse event (irAE) and irAE groups.

	All	irAE (*n* = 13)	no-irAE (*n* = 10)	*p* Value
V	70 (45–87)	68	73 (45–80)	0.99
Sex				
Male	18	9	9	0.33
Female	5	4	1	
Histology				
Clear cell carcinoma	19	11	8	0.99
Other	4	2	2	
Times of CCB therapy				
1–2	8	4	4	0.65
3–4	15	9	6	
Effect of CCB therapy				
PD, SD	9	5	4	0.48
PR, CR	13	8	5	
No evaluation	1	0	1	

PD: Progression disease, SD: Stable disease, PR: Partial response, CR: Complete response.

## Data Availability

Not applicable.

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
