# Peer review of "CD21^lo^ B Cells Could Be a Potential Predictor of Immune-Related Adverse Events in Renal Cell Carcinoma"

_jpm, 2022, doi:10.3390/jpm12060888_

Round 1
Reviewer 1 Report
The authors describe the potential of CD21lo B cells as a predictor of adverse events in renal cell carcinoma after immune check point therapy. The study is an interesting one with potential implications in assessing patients at risk of developing adverse events after check point therapy. There are however a few minor concerns that must be addressed.
1. The title should be corrected for typos.
2. In figure 1, the figure numbering is not clear.
3. On lines 141 and 142, the authors mention “Therefore, ICIs may impact B cell function indirectly by exerting effects on T cells.” This discussion should be further elaborated and backed by some references.
Author Response
The following paragraphs changed to improve the introduction.
(On lines 41 and 51)
The prediction and management of irAEs are important in clinical practice. Biomarker for early prediction of irAEs have been explored [4-12]. In case of non-organ specific biomarkers in CCB therapy, Das et al. reported that among patients with melanoma who received CCB therapy, the proportion of B cells in the peripheral blood decreases, whereas that of CD21lo B cells increases, after treatment in patients who developed grade 3 or higher irAEs. This finding suggests that early changes in B cells following CCB therapy can be used to identify patients who are at increased risk of irAEs [4]. However, as the dose of CCB therapy is different for RCC and melanoma, the association between early B cell changes and irAEs in patients with RCC being treated with CCB therapy remains unclear. Since the incidence of irAEs varies depending on the dose of ICIs [3].
- The title should be corrected for typos.
Reply: Thanks for pointing out my mistake. A typo has been corrected.
- In figure 1, the figure numbering is not clear.
Reply: Thank you for your advice. Figure 1 corrected.
- On lines 141 and 142, the authors mention “Therefore, ICIs may impact B cell function indirectly by exerting effects on T cells.” This discussion should be further elaborated and backed by some references.
Reply: The following sentence is included in the discussion for explanation.
(On lines 141 and 146)
LAG-3 CD4+ T cells increased by ICIs selectively activate Regulatory T cells to relate both anti-cancer effect and autoimmune disease [135]. In addition, LAG-3 CD4+ T cells bind to MHC class II molecules on the surface of antigen presenting cells (APCs) [146]. Next, APCs induce differentiation into Th1/Th2 cells. Th1 cells then activate B cells [157].

Reviewer 2 Report
The article titled "CD21lo B cells coul a potential predictor of immune-related adverse events in renal cell carcinoma" by Nishimura eta l., is an interesting article that touches on an important aspect on RCC treatment using immune checkpoint blocking agents.
There are two concerns regarding this manuscript:
1) The discussion needs to be improved.
2) The authors have used a very limited list of references, There are a plethora of literature on similar topics,; therefore, the authors need to expand on the cited literature to adequately and effectively cover the topics presented in their manuscript.
Author Response
The following paragraphs changed to improve the introduction.
(On lines 41 and 51)
The prediction and management of irAEs are important in clinical practice. Biomarker for early prediction of irAEs have been explored [4-12]. In case of non-organ specific biomarkers in CCB therapy, Das et al. reported that among patients with melanoma who received CCB therapy, the proportion of B cells in the peripheral blood decreases, whereas that of CD21lo B cells increases, after treatment in patients who developed grade 3 or higher irAEs. This finding suggests that early changes in B cells following CCB therapy can be used to identify patients who are at increased risk of irAEs [4]. However, as the dose of CCB therapy is different for RCC and melanoma, the association between early B cell changes and irAEs in patients with RCC being treated with CCB therapy remains unclear. Since the incidence of irAEs varies depending on the dose of ICIs [3].
1) The discussion needs to be improved.
Reply: The following sentence is included in the discussion for explanation.
(On lines 141 and 146)
LAG-3 CD4+ T cells increased by ICIs selectively activate Regulatory T cells to relate both anti-cancer effect and autoimmune disease [135]. In addition, LAG-3 CD4+ T cells bind to MHC class II molecules on the surface of antigen presenting cells (APCs) [146]. Next, APCs induce differentiation into Th1/Th2 cells. Th1 cells then activate B cells [157].
2) The authors have used a very limited list of references, There are a plethora of literature on similar topics,; therefore, the authors need to expand on the cited literature to adequately and effectively cover the topics presented in their manuscript.
Reply: Thank you for your advice. Added references to the introduction and discussion.
